# Determination of 20 Endocrine-Disrupting Compounds in the Buffalo Milk Production Chain and Commercial Bovine Milk by UHPLC–MS/MS and HPLC–FLD

**DOI:** 10.3390/ani12040410

**Published:** 2022-02-09

**Authors:** Ilaria Di Marco Pisciottano, Grazia Guadagnuolo, Fabio Busico, Luca Alessandroni, Bruno Neri, Domenico Vecchio, Gabriele Di Vuolo, Giovanna Cappelli, Alessandra Martucciello, Pasquale Gallo

**Affiliations:** 1Istituto Zooprofilattico Sperimentale del Mezzogiorno, Department of Chemistry, via Salute 2, 80055 Portici, Italy; grazia.guadagnuolo@izsmportici.it (G.G.); pasquale.gallo@izsmportici.it (P.G.); 2Istituto Zooprofilattico Sperimentale del Lazio e della Toscana, Department of Chemistry, via Appia Nuova 1411, 00178 Rome, Italy; fabio.busico@izslt.it (F.B.); luca.alessandroni@izslt.it (L.A.); bruno.neri@izslt.it (B.N.); 3Istituto Zooprofilattico Sperimentale del Mezzogiorno, National Reference Centre for Hygiene and Technologies of Water Buffalo Farming and Productions (CReNBuf), via Salute 2, 80055 Portici, Italy; domenico.vecchio@izsmportici.it (D.V.); gabriele.divuolo@izsmportici.it (G.D.V.); giovanna.cappelli@izsmportici.it (G.C.); alessandra.martucciello@izsmportici.it (A.M.)

**Keywords:** BPA, endocrine disruptors, bisphenols, buffalo milk, animal feed

## Abstract

**Simple Summary:**

The restriction of the use of bisphenol A (BPA) in industrial production has led manufacturers to employ several BPA analogues. The endocrine-disrupting activity of these substitutes has been proven, and some of them are already widespread in the environment. The intake of these endocrine-disrupting chemicals through diet represents a public concern, and food contamination data are still scarce in scientific literature. To investigate the levels of BPA and its analogues in the production chain of buffalo milk, we developed and validated two analytical methods based on UHPLC–MS/MS and HPLC–FLD. The methods were used to analyse feed and feed additives, raw milk, drinking water, and blood serum samples from buffalo farms in the Campania region, Southern Italy. BPA was found to be the most abundant contaminant, proving that the presence of this compound is still significant and should be monitored.

**Abstract:**

Bisphenol A (BPA) and some of its analogues are well known as endocrine-disrupting chemicals (EDCs), i.e., compounds that can affect the physiological hormonal pathways in both animals and humans, causing adverse health effects. The intake of these substances through diet represents a public concern, compounded by the scarce data in the literature about contamination levels in food. In the framework of a research project, funded by the Italian Ministry of Health, we determined the contamination levels of BPA and 19 EDCs in the production chain of buffalo milk, analysing feed, drinking water, buffalo milk, and blood sera. Overall, 201 feed, 9 feed additive, 62 drinking water, 46 milk, and 190 blood serum samples were collected from 10 buffalo farms in the Campania region, Southern Italy, between 2019 and 2020, and analysed. Moreover, 15 retail bovine milk samples packaged both in Tetra Pak and in PET were analysed to further evaluate consumers’ exposure to EDCs. The results of our work showed no contamination by EDCs in drinking water samples, whereas in 43% of all of the other samples from the farms at least one bisphenol was detected. The most abundant bisphenol detected was BPA (32% of the samples from the farms and 80% of the retail milk samples), thus proving that this compound is still widely used for plastic production.

## 1. Introduction

Bisphenol A (BPA) is an endocrine-disrupting chemical (EDC) employed in the industrial production of several materials, such as polycarbonate plastics and epoxy resin inner linings for metal food and beverage cans. BPA is one of the most ubiquitous environmental contaminants, having been found even in indoor dust [1,2,3,4,5]; therefore, it represents a public health concern. Following the restrictions on the use of BPA placed by the European Commission and some European Union member states [6,7,8], manufacturers started to replace BPA with some of its analogues. Unfortunately, these substitutes have physicochemical properties similar to those of BPA and, for some of them, relevant endocrine-disrupting activity has been proven [9,10,11,12,13]. There are many human exposure routes for BPA and EDCs in general, including occupational exposure, the ingestion of contaminated food, inhalation, and dermal absorption from the environment [14,15,16,17]. In 2015, the European Food Safety Authority (EFSA) published a scientific opinion about the risk to human health posed by the occurrence of BPA in foodstuffs [18]; although it was found that BPA intake via diet does not represent a risk to human health, the panel identified food as the main source of intake. Food contamination can occur through migration of EDCs from packaging, but it can also be a consequence of environmental pollution. For example, EDCs can transfer from the polluted environment to the raw materials used as food ingredients, or from contaminated soil, water, or atmospheric deposition from nearby industrial activity [19,20]. Moreover, animals fed with contaminated feed can bioaccumulate EDCs in their adipose tissue, because of their good lipophilicity, and then can eliminate them through major excretion routes, including milk [21,22,23]. Additionally, all storage containers and processing machines used along the production chain can be sources of contamination if they are made with materials that can leak EDCs into the food in contact with them. Whereas the presence of BPA and other bisphenols in breast milk has been widely studied, there are scarce data about bisphenol contamination in bovine milk [24,25,26,27], and no data at all about buffalo milk. Moreover, even though some researchers have studied bisphenol concentrations in feed [28,29,30,31], none have simultaneously analysed bovine milk, sera, and drinking water from the same farms. Therefore, a comprehensive contamination assessment of both the bovine and buffalo milk production chains has never been performed before. To study human exposure to these compounds, in the framework of a research project funded by the Italian Ministry of Health, we investigated possible contamination levels from many EDCs in the production cycle of buffalo milk. A novel testing method using liquid chromatography coupled with tandem mass spectrometry (UHPLC–MS/MS), previously developed and validated in our laboratory, was used for the determination of 17 bisphenols in drinking water, feed, and feed additives, as well as in raw buffalo milk and blood sera. A second testing method, based on liquid chromatography coupled with fluorescence detection (HPLC–FLD), was also developed and validated to determine 16 EDCs in milk samples. The HPLC–FLD method allowed us to quantify down to 1.0 ng/g in milk for most of the tested EDCs and is a reliable alternative method to the confirmatory UHPLC–MS/MS analysis of milk. Moreover, HPLC–FLD was also able to determine three alkylphenols (4-nonylphenol, 4-octylphenol, and 4-tert-octylphenol) not detected by the UHPLC–MS/MS method. The determination via liquid chromatography coupled with tandem mass spectrometry allowed us to unambiguously identify and quantify the EDCs down to 0.010 ng/g in serum. Both methods were used to analyse 46 milk samples, whereas only the UHPLC–MS/MS method was used to analyse 201 feed, 9 feed additive, 62 drinking water, and 190 blood serum samples collected from 10 buffalo farms in the Campania region (Southern Italy) during 2019 and 2020. These samples were collected within the monitoring plan of several buffalo farms, in order to study the presence of bisphenols in the production chain of milk destined to make mozzarella cheese. Furthermore, 15 retail bovine milk samples packaged both in Tetra Pak and in PET, collected from the markets in the Campania and Lazio regions, were analysed in order to evaluate the migration of EDCs from packaging. The results of these monitoring activities are reported herein.

## 2. Materials and Methods

### 2.1. Chemicals and Standard Reference Materials

The standard reference materials for the bisphenols (BPs) and internal standards (ISs) were supplied by Aldrich (Sigma-Aldrich, Milano, Italy), and were all of analytical purity grade. HPLC-grade methanol (MeOH) and glacial acetic acid (97% *v*/*v*, purity grade) were obtained from VWR International (Radnor, PA, USA). HPLC-grade acetonitrile (ACN) and ammonium acetate (analytical purity grade) were purchased from Carlo Erba (Milan, Italy). HPLC-grade water was produced in-house using a Milli-Q laboratory system (Millipore, Bedford, MA, USA). To avoid any possible contamination deriving from plastic materials, only glassware, previously rinsed twice with both acetonitrile and methanol, was used during analysis.

### 2.2. Sample Collection and Storage

The samples tested were collected from 10 buffalo farms in the Campania region, over the period 2019–2020; for each farm, 4–7 samples were taken during different periods and buffalo lactation phases. A total of 201 feed (raw materials, compound, and complete feed, unifeed for buffaloes in both dry and lactation phases), 9 feed additive, 62 drinking water (for buffaloes in both dry and lactation phases), 46 raw milk, and 190 blood serum samples were collected from the selected farms and analysed. Blood samples were collected twice from nine farms and once from one farm, during different periods and lactation phases. Furthermore, buffalo blood samples were tested for some haematochemical parameters related to wellbeing and to the metabolism of each selected animal. All samples were collected in glass or jute containers in order to avoid any contact with plastics, except for those already present in the milk production cycle steps; in this way, undesired contamination by EDCs during sampling was avoided. Furthermore, 15 samples of retail bovine milk, packaged both in PET and in Tetra Pak containers, were collected from markets in both the Campania and Lazio regions (10 from Campania and 5 from Lazio) and analysed to evaluate consumers’ possible exposure to EDCs due to environmental pollution or to migration from the packaging.

The more perishable milk and blood serum samples were stored at temperatures ≤ −18 °C, whereas drinking water samples were kept at 5 °C ± 3 °C. Feed and feed additives were homogenised using a knife mill and stored either at room temperature or at 5 °C ± 3 °C, depending on their water content; all of the dry materials were kept in a dry and dark place at room temperature, while the moist ones were stored at 5 °C ± 3 °C.

### 2.3. Sample Purification and UHPLC–MS/MS Analysis

Blood serum samples (3 mL) were incubated for 3 h at 37 °C in a thermostatic water bath, adding 60 µL of β-glucuronidase/arylsulfatase from *Helix pomatia*, diluted 10-fold with HPLC-grade water, for deconjugation before analysis of haematochemical parameters; then, the samples underwent the test for determination of bisphenols. The UHPLC–MS/MS method used for the monitoring activities we describe hereafter was previously developed and validated in-house in our laboratory (data submitted). Drinking water and blood serum clean-up was performed by liquid partition using organic solvents (methanol and acetonitrile, respectively). Feed, feed additives, and milk samples were cleaned up using SPE cartridges with molecularly imprinted polymers (MIPs) specific for BPA; in a previous work, we proved that they are also effective for several BPA analogues [32,33,34]. The chromatographic separation of the analytes was performed on a reversed-phase (RP) column with a hexyl-phenyl stationary phase, applying a linear gradient elution using water and methanol buffered with ammonium acetate and acetic acid as mobile phases. An Exion UHPLC chromatograph (SCIEX)—composed of a quaternary pump system, an autosampler, and a thermostatic oven for the column—was used. The mass spectrometry detector coupled with the UHPLC—a QTRAP 6500^+^ system (SCIEX)—was run in multiple-reaction monitoring (MRM), both in positive and in negative ionisation mode. For each EDC, the two most abundant MRM transitions were selected to monitor the quantifier and qualifier ions, allowing us to unambiguously identify and quantify all of the analytes. The quantification was carried out using linear regression internal standard (IS) calibration curves, calculated by plotting the EDC-to-IS chromatographic peak area ratio vs. the standard solution concentrations. Method validation was carried out at three concentration levels for drinking water, blood serum, and milk, and at two levels for feed and feed additives, performing at least 4 replicates for each validation level and each matrix, over a minimum of 2 working sessions. Afterwards, for quality control of method trueness, a blank sample for each matrix spiked with the 17 bisphenols was run during each working session. Moreover, during each working session, a process blank consisting of the extraction solvent was cleaned up and analysed in order to evaluate possible interference from both materials (e.g., MIP cartridges, reagents, solvents, glassware) and the environment (e.g., air, dust). 

### 2.4. Sample Purification and HPLC–FLD Analysis

The raw milk samples analysed by HPLC–FLD were cleaned up using MIP cartridges, but only after an additional purification step consisting of a dispersive SPE (dSPE) with QuEChERS. The chromatographic separation of the EDCs was performed on a reversed-phase (RP) column with a polar-embedded C18 stationary phase, applying a linear gradient elution using water and acetonitrile as mobile phases. The fluorescence detector was set at 230 nm and 315 nm as excitation and emission wavelengths, respectively. The quantification was carried out using linear regression external calibration curves, calculated by plotting the EDCs’ chromatographic peak areas vs. the standard solution concentrations. Method validation was carried out at 3 concentration levels, performing a total of 12 replicates for each level over 2 working sessions. During sample analysis, a blank sample was spiked with all 16 EDCs, cleaned up, and analysed for quality control of the method trueness of the working session. Moreover, during each working session, a process blank was tested to evaluate possible interference from both materials (e.g., SPE cartridges, reagents, solvents, glassware) and the environment (e.g., air, dust).

Table 1 shows all of the EDCs analysed, including the internal standards used for the UHPLC–MS/MS determination, along with the indication of the method used to detect the compounds.

### 2.5. Method Validation

Both of the test methods were validated in-house according to Regulation (EU) N. 2017/625 [35], evaluating analytical performance parameters such as specificity, trueness, precision, linearity of the detector response, ruggedness for slight variations, relative expanded measurement uncertainty, the limits of quantification (LOQs), and the limits of detection (LODs). The UHPLC–MS/MS method was also validated according to the Commission Decision (EC) N. 2002/657 [36] requirements concerning the criteria for unambiguous identification using mass spectrometry analysis (data submitted). Method specificity was assessed by analysing samples not contaminated by the EDCs and verifying that there were no interfering peaks in the diagnostic areas of the chromatograms. The trueness and precision were evaluated for each matrix as percentage mean recoveries and percentage coefficients of variation, respectively, spiking blank samples at the chosen concentration levels. Standard calibration curves, consisting of at least 4 points in the concentration range 0.1–500.0 ng/mL, were calculated during each working session in both solvents and matrix-matched for all of the EDCs. The linearity of the detector response, expressed as a coefficient of determination (R^2^), was verified to be always higher than 0.95 for all of the EDCs and matrices. The ruggedness was evaluated by introducing slight variations in the method (different batches of solvents and reagents, different analysts, different room temperatures, etc.) and verifying that none of them affected the analytical performance in terms of trueness and precision. The relative expanded measurement uncertainty was calculated via a metrological approach, considering the combined contributions of calibration, repeatability, weight and volume measurements, recovery, and the reference materials, using a coverage factor k = 2 and 95% probability. The LOQs of each EDC and in each matrix were determined by spiking blank samples at decreasing concentrations of all of the analytes for the UHPLC–MS/MS method; conversely, for the HPLC–FLD method, the LOQs were calculated as 10 times the standard deviation from the mean bias signal of 10 blank samples, at the retention time of each analyte. All of the performance parameters calculated for the validation of both methods were satisfactory in all matrices. For the HPLC–FLD method, the LOQ for all of the EDCs was 1.0 ng/g, except for bisphenol BP, showing the LOQ at 3.0 ng/g; on the other hand, for the UHPLC–MS/MS method, the LOQs were related to the EDC and the matrix. Specifically, the LOQs of the 17 EDCs in UHPLC–MS/MS were in the concentration range 0.01–1.00 ng/mL for drinking water, 0.01–1.00 ng/g for blood serum, 0.1–5.0 ng/mL for milk, and 1.0–10.0 ng/g for feed and feed additive samples. The LODs were calculated from the LOQs according to the equation LOD = LOQ/3.3 for the UHPLC–MS/MS method, and as 3 times the standard deviation calculated from 10 blank samples for the HPLC–FLD method. Following that, the LOD of the HPLC–FLD method was 0.2 ng/g for all of the EDCs, except for the BPBP, which had an LOD of 0.6 ng/g. For the UHPLC–MS/MS method, the LODs of the 17 EDCs were in the range 0.003–0.30 ng/mL for drinking water, 0.003–0.30 ng/mL for blood serum, 0.03–1.5 ng/mL for milk, and 0.3–3.0 ng/g for feed and feed additive samples.

## 3. Results

The main purpose of this study was to investigate the possible presence of BPA and its analogues in some buffalo breeding farms in Southern Italy. Buffalo milk is a remarkable product because it is used to produce mozzarella cheese, which represents a relevant economic interest. The UHPLC–MS/MS method was the main method used for this purpose. A total of 201 feed (raw materials, compound, and complete feed, unifeed for buffaloes in both dry and lactation phases), 9 feed additive, 62 drinking water (for buffaloes in both dry and lactation phases), 46 raw milk, and 190 blood serum samples were collected and analysed. 

At the same time, the 46 raw milk and 9 drinking water samples were also analysed via the HPLC–FLD method, in order to compare results and, above all, to obtain data about the possible presence of 4-octylphenol (4-OP), 4-tert-octylphenol (4-t-OP), and 4-nonylphenol (4-NP).

Furthermore, 10 retail milk samples were collected from markets in the Campania region and analysed by UHPLC–MS/MS; similarly, 5 retail milk samples were collected in the Lazio region, and then analysed by HPLC–FLD. 

### 3.1. Samples Collected from the Buffalo Breeding Farms in the Campania Region

The UHPLC–MS/MS analysis of the samples collected from the 10 selected buffalo farms in the Campania region showed that the most abundant EDC in all of the tested matrices was BPA. In fact, ~30% of the analysed samples were contaminated by BPA, at concentrations ranging from 0.16 ng/g to 174.7 ng/g. The second most detected bisphenol was bisphenol F (BPF), identified in 3.5% of the samples, in the concentration range 0.5–142.2 ng/g. Among all of the studied bisphenols, only BADGE, BFDGE, BPS, BPAF, and BPE were sporadically detected (rates between 0.6 and 1.4% of all the samples). Furthermore, we observed no EDC contamination in drinking water samples.

Table 2 shows the contamination ranges, along with the mean and median values of the EDCs determined by UHPLC–MS/MS—expressed as ng/g for blood serum, feed, and feed additives, and as ng/mL for raw milk—and the frequency of detection, expressed as the percentage of positive samples, grouped by matrix. As can be seen, only BPA, BPF, BADGE, BFDGE, BPS, BPAF, and BPE were identified in at least one of the samples tested; moreover, the drinking water samples were not contaminated by any bisphenol.

It is noteworthy that most of the feed and feed additives were contaminated—mainly by BPA (54.2% and 88.9%, respectively)—in a wide concentration range, up to 174.7 ng/g; they were contaminated at a minor rate by BPF (4.5% and 11.1%, respectively), BPE, BFDGE, and BPS. In particular, BPE was identified in 33.3% of feed additives, although its concentrations were quite low (up to 10.0 ng/g). On the other hand, BFDGE and BPS were identified only in feed samples (3.0% and 2.5%, respectively), up to 40.3 ng/g. The group of Wang et al. [29,31] determined the concentrations of 12 bisphenols in 20 and 203 feed samples, and they found that BPA was the most abundant contaminant (0.51–36.86 ng/g), followed by BPS (0.03–34.93 ng/g), BPF (0.20–54.12 ng/g), BPAF (0.08–1.57 ng/g), and BPAP (0.50–2.82 ng/g). Except for BPAF and BPAP, which were detected in only a few samples anyway (2 and 1 out of 223, respectively), the results of Wang’s studies are consistent with our results, showing that BPS and BPF are the most abundant bisphenols in feed other than BPA. Although only a few samples of feed additives were tested (*n* = 9), the wide contamination range detected suggests the need for more data. On the other hand, the large number of feed samples (*n* = 201) accounts for a good estimation of bisphenol contamination entering the buffalo milk production chain. We suppose that the broad presence of bisphenols in feed and feed additives derives from packaging materials made of plastics, rather than from the environment; in this case, it is reasonable to find mainly BPA and its most common substitutes in plastic manufacturing. The contamination levels seem relatively low, but not negligible, especially regarding BPA; it should be considered that a daily feeding ration of approximately 20 kg, using a feed contaminated at 100 ng/g of BPA, is equivalent to an intake for the animal of 100 µg/kg × 20 kg = 2000 µg, corresponding to 2 mg/day.

If the feed is contaminated, the consequential concern is about buffalo milk. Usually, feeding is the main route of contamination; it is therefore not surprising that raw buffalo milk is also contaminated by bisphenols. A significant correlation between polluted feed and bisphenols in raw milk was observed for BPA and BPF (58.7% and 17.4% of samples, respectively), but in all cases we found bisphenol concentrations below 10 ng/mL. Santonicola et al. [25] analysed 72 milk samples collected from cow farms via different milking techniques and found BPA in the concentration range 0.035–2.776 µg/L; no BPA was found in feed samples collected monthly for 4 months from the same farm. Mercogliano et al. [27] also investigated BPA levels in 92 cow milk samples collected from a dairy company at different steps of the production chain. BPA was identified in 35% of the samples, at between 0.1 and 2.833 µg/L, showing results consistent with the levels found by Santonicola et al. [25], but slightly lower than our results in milk samples. The group of Santonicola et al. [26] also studied BPF contamination in 84 cow milk samples collected from a dairy company at various levels of the production chain. Their results showed that more than 50% of the cow milk samples were contaminated by BPF at concentrations ranging from 0.1 to 2.686 µg/L; in this case, the concentration levels were also slightly lower than those we determined. 

BPA, BADGE, BFDGE, and BPAF were also detected in a few samples of buffalo blood serum (in 3.7%, 3.7%, 0.5%, and 1.6% of 190 samples, respectively). It seems there is no correlation between bisphenols’ relative abundance and contamination rates in feed; moreover, the concentrations found were very low. The results suggest that the contaminants in blood could have an environmental origin, considering that in buffalo breeding farms the animals are usually free for grazing.

The buffalo blood samples were also tested for some haematochemical parameters related to wellbeing and to the metabolism of each selected animal; the results (not reported) did not show statistically significant differences for the same production phase and between the various farms.

For BPA, which was the most abundant contaminant, a study of the distribution in the feed samples was carried out; the results, reported in Figure 1, showed that there was no significant difference in the BPA distribution with respect to either the farms or the concentration levels. 

Overall, in around half of the samples from each farm, BPA was not detected at all, whereas most of the feed contained BPA at between 1.0 ng/g and 50.0 ng/g, regardless of the farm. On the other hand, the samples from two farms were contaminated by BPA at concentrations ranging from 50.0 ng/g to 100.0 ng/g, and in only one farm was BPA found at levels higher than 100.0 ng/g. Considering this, our results are mainly consistent with the BPA contamination levels determined in feed by Wang et al. [29,31], in the ranges 0.51–36.86 ng/g and 0.92–7.77 µg/kg, respectively. Moreover, the group of Xiong et al. [28] analysed the presence of 9 bisphenols in 30 feed samples, and found BPA in only 1 sample, at 12.60 µg/kg, whereas the other bisphenols were not found at all. Furthermore, Wang et al. [30] analysed 30 samples of plastic packaging for feed, and found that all of the samples were contaminated by BPA, at concentrations ranging from 2.91 ng/g to 8575 ng/g. Other bisphenols were identified by Wang et al. [31] in the plastic packaging, including BPS (86.7% of the samples), BPF (36.7%), and BPAF (13.3%); furthermore, they performed migration experiments that confirmed that BPA transfers from packaging into solid feed. However, the results concerning the presence of bisphenols in packaging are consistent with the overall bisphenol contamination profile observed in feed samples by both our group and that of Wang et al.

The HPLC–FLD analysis of 46 raw buffalo milk samples, reported in Table 3, showed the presence of 4-t-OP and of BFDGE above the LOQs in one (1.41 ng/g) and two (1.10 and 1.33 ng/g) samples, respectively; apart from these, BPF, BPC, and 4-NP were also detected in the analysed samples at concentrations between their LODs and the LOQs. The results concerning BFDGE and BPF were not confirmed by the UHPLC–MS/MS analysis, whereas BPC was identified at a concentration lower than its LOD by the UHPLC–MS/MS method and, therefore, could not be confirmed unambiguously.

### 3.2. Retail Milk Samples from the Lazio and Campania Regions

To evaluate consumers’ exposure to EDCs from retail bovine milk as well, 15 samples, packaged both in Tetra Pak and in PET, were collected from markets of Campania and Lazio. In detail, 10 milk samples were collected from Campania and analysed using the UHPLC–MS/MS method, while 5 milk samples collected from Lazio were analysed by HPLC–FLD. The results of the UHPLC–MS/MS analysis (Table 4) confirmed that BPA is the most detected contaminant among the studied bisphenols; moreover, BPF was the only other bisphenol quantified, thus confirming the contamination profile already observed in the analysis of buffalo milk samples. The contamination profile in retail milk was quite similar to that observed by Grumetto et al. [24], but the bisphenol concentrations they quantified were much higher than ours. In fact, Grumetto et al. determined BPA, BPF, and BPB in 68 retail milk samples at concentrations up to 521, 67, and 26.2 ng/mL, respectively; BADGE and BFDGE were not determined at all. The five milk samples from Lazio (Table 5) confirmed the ubiquitous presence of BPA above the LOQ, along with the detection of 4-t-OP, 4-OP, and the sum of BPP and BPM (not chromatographically resolved) at concentrations between the LOD and the LOQ. The small number of samples did not allow us to perform a statistically significant evaluation, but apparently there was no correlation between the packaging materials and the BPA levels. 

### 3.3. Risk Assessment for Consumers’ Health

On the basis of the contamination data obtained from the raw buffalo milk samples, a calculation of consumers’ possible intake of BPA through consumption was carried out. To this end, the temporary tolerable daily intake (t-TDI) established by the EFSA in 2015 was used; lacking reference values concerning buffalo milk consumption, the data reported for consumption of bovine milk and derived products from the WHO were employed.

Both the best case and worst case scenarios were calculated using the lowest and the highest concentrations of BPA determined in raw buffalo milk, respectively. The consumers’ intake of BPA, expressed as percentage of the t-TDI, was calculated as follows:

Temporary TDI (t-TDI) for the BPA [18]: 4 µg/kg bw/day;

Consumption of cow milk and milk products for WHO-EU Europe cluster countries [37]: 4.78333 g/kg bw/day;

Best case scenario: BPA concentration at 0.5 ng/g (minimum detected level):

4.78333 g/kg bw/day × 0.5 ng/g × 10^−3^ = 0.002391665 µg/kg bw/day (0.06% of the t-TDI);

Worst case scenario: BPA concentration at 5.6 ng/g (maximum detected level):

4.78333 g/kg bw/day × 5.6 ng/g × 10^−3^ = 0.026786648 µg/kg bw/day (0.7% of the t-TDI).

The results of our calculations indicate a very low risk to consumers’ health due to the consumption of milk and derived products; these results are the best approximation regarding our results.

## 4. Discussion

The presence of EDCs in food can be the result of several factors, such as environmental pollution, contamination during the production process, or the migration from packaging materials in direct contact with the food itself. BPA is a ubiquitous contaminant, and its detrimental effects to both human and animal health are well known and scientifically proven. Therefore, the presence of this chemical in food could represent a serious risk to human health—especially for some groups of susceptible individuals, such as toddlers, children, and pregnant women.

This study aimed to investigate for the first time, to our knowledge, the presence of EDCs along the production chain of buffalo milk. The results showed that feed is the main source of contamination by bisphenols in the production chain of buffalo milk.

Overall, our results confirmed the ubiquitous contamination by BPA, which was consequently the most abundant EDC in feed and in milk, followed by BPF, BPE, BADGE, BFDGE, BPAF, and BPS.

BPA was detected in raw buffalo milk samples (58.7% of positive samples) from all of the farms, at concentration levels ranging from 0.5 to 5.6 ng/mL. The contamination levels were relatively low; therefore, there is no evidence that the BPA intake from buffalo milk represents a significant risk to human health. Moreover, the risk assessment calculated by considering the highest BPA concentration determined in contaminated raw buffalo milk—the worst case scenario—showed that its intake would be only 0.7% of the t-TDI set by the EFSA. Because of the lack of more specific data, the risk evaluation was carried out considering the consumption data indicated by the WHO for bovine milk and derived products. No correlation was observed between the lactation phases of the buffaloes and the contamination levels of the EDCs, and no statistically significant variations were observed in the haematochemical parameters of the animals with respect to the presence of EDCs in the milk. BPA and BPF were the most relevant bisphenols entering the buffalo milk production chain; their respective distribution profiles in feed and feed additives were very similar to those observed in milk, thus being evidence of the possible transfer of contaminants from feed to food along the production chain.

Although only a few samples were tested, it is interesting to note that BPA and BPF were also detected in commercially available retail bovine milk; the concentrations we determined were of the same magnitude observed in raw buffalo milk, as were the percentages of contaminated samples, and no relationship could be found with the Tetra Pak or PET material. For these reasons, we cannot state that bisphenols derive from packaging—it is more probable the milk itself is contaminated.

## 5. Conclusions

The results of this work showed undeniable contamination in the production chain of buffalo milk due to EDCs—and in particular to BPA and BPF—with the feed being the major contamination source. There is evidence of the transfer of the EDCs from the feed to the milk. Although the contamination levels are very low, we are studying a model to predict the transfer of BPA from buffalo milk to the derived mozzarella cheese, in order to evaluate the possible intake by consumers.

Similarly, in some retail bovine milk, low concentrations of BPA and BPF were detected; their presence was independent of the material used for the packaging—more likely it depends on the contamination of the milk itself.

## Figures and Tables

**Figure 1 animals-12-00410-f001:**
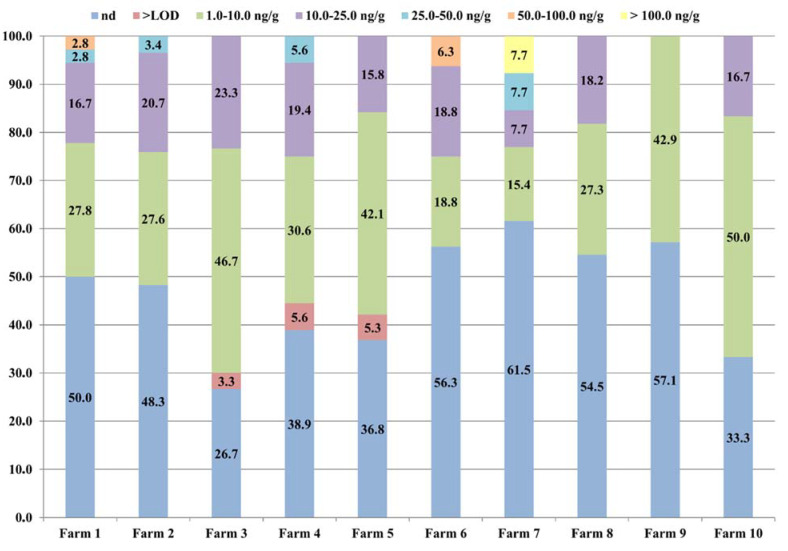
The BPA distribution (%) in animal feed samples with respect to the 10 buffalo farms monitored. The number on each stack indicates the percentage of samples belonging to the respective concentration range. In the group “nd” are reported all of the samples with a BPA concentration lower than the LOD, whereas the group “>LOD” indicates the samples containing BPA at levels between the LOD and the LOQ (LOQ = 1.0 ng/g; LOD = 0.3 ng/g).

**Table 1 animals-12-00410-t001:** List of the EDCs studied with the abbreviation, the CAS number, and the indication of the method used to detect each compound.

EDC (Abbreviation)	CAS Number	UHPLC–MS/MS	HPLC–FLD
Bisphenol A (BPA)	80-05-7	X	X
Bisphenol A diglycidyl ether (BADGE)	1675-54-3	X	X
Bisphenol AF (BPAF)	1478-61-1	X	X
Bisphenol AP (BPAP)	1571-75-1	X	X
Bisphenol B (BPB)	77-40-7	X	X
Bisphenol BP (BPBP)	1844-01-5	X	X
Bisphenol C (BPC)	79-97-0	X	X
Bisphenol E (BPE)	2081-08-5	X	X
Bisphenol F (BPF)	620-92-8	X	X
Bisphenol F diglycidyl ether (BFDGE)	2095-03-6	X	X
Bisphenol G (BPG)	127-54-8	X	-
Bisphenol M (BPM)	13595-25-0	X	X
Bisphenol P (BPP)	2167-51-3	X	X
Bisphenol PH (BPPH)	24038-68-4	X	-
Bisphenol S (BPS)	80-09-1	X	-
Bisphenol TMC (BPTMC)	129188-99-4	X	-
Bisphenol Z (BPZ)	843-55-0	X	X
4-Octylphenol (4-OP)	1806-26-4	-	X
4-tert-Octylphenol (4-t-OP)	140-66-9	-	X
4-Nonylphenol (4-NP)	104-40-5	-	X
Bisphenol C2 (BPC2) *	14868-03-2	X	-
Bisphenol A-d_16_ (BPA-d_16_) *	96210-87-6	X	-

* Internal standards for the UHPLC–MS/MS analysis.

**Table 2 animals-12-00410-t002:** The concentration ranges, the percentage of positive samples, and the mean and median values of the EDCs determined in the samples by UHPLC–MS/MS, grouped by matrix. No EDCs were detected in any of the 62 drinking water samples. (*nd* = not detected).

		Feed, ng/g (*n* = 201)	Feed Additives, ng/g (*n* = 9)	Raw Milk, ng/mL (*n* = 46)	Blood Serum, ng/g (*n* = 190)
BPA	Range	1.2–174.7 (54.2%)	4.1–44.2 (88.9%)	0.5–5.6 (58.7%)	0.16–6.39 (3.7%)
	Mean	12.5 ng/g	18.3 ng/g	1.4 ng/mL	1.43 ng/g
	Median	7.9 ng/g	16.0 ng/g	0.9 ng/mL	0.47 ng/g
	LOQ	1.0 ng/g	1.0 ng/g	0.5 ng/mL	0.10 ng/g
	LOD	0.3 ng/g	0.3 ng/g	0.2 ng/mL	0.03 ng/g
BPF	Range	10.9–142.2 (4.5%)	26.2 (11.1%)	0.5–8.7 (17.4%)	*nd*
	Mean	37.8 ng/g	26.2 ng/g	3.1 ng/g	*-*
	Median	15.2 ng/g	26.2 ng/g	1.6 ng/g	*-*
	LOQ	10.0 ng/g	10.0 ng/g	0.5 ng/mL	0.20 ng/g
	LOD	3.0 ng/g	3.0 ng/g	0.2 ng/mL	0.06 ng/g
BADGE	Range	*nd*	*nd*	*nd*	0.07–0.23 (3.7%)
	Mean	*-*	*-*	*-*	0.12 ng/g
	Median	*-*	*-*	*-*	0.11 ng/g
	LOQ	1.0 ng/g	1.0 ng/g	0.1 ng/mL	0.05 ng/g
	LOD	0.3 ng/g	0.3 ng/g	0.03 ng/mL	0.02 ng/g
BFDGE	Range	5.4–40.3 (3.0%)	*nd*	*nd*	0.49 (0.5%)
	Mean	18.2 ng/g	*-*	*-*	0.49 ng/g
	Median	14.1 ng/g	*-*	*-*	0.49 ng/g
	LOQ	5.0 ng/g	5.0 ng/g	0.5 ng/mL	0.20 ng/g
	LOD	1.5 ng/g	1.5 ng/g	0.2 ng/mL	0.06 ng/g
BPS	Range	1.2–7.4 (2.5%)	*nd*	*nd*	*nd*
	Mean	4.2 ng/g	*-*	*-*	*-*
	Median	5.0 ng/g	*-*	*-*	*-*
	LOQ	1.0 ng/g	1.0 ng/g	5.0 ng/mL *	0.010 ng/g
	LOD	0.3 ng/g	0.3 ng/g	1.5 ng/mL	0.003 ng/g
BPAF	Range	*nd*	*nd*	3.0 (2.2%)	0.14–1.16 (1.6%)
	Mean	*-*	*-*	3.0 ng/mL	0.52 ng/g
	Median	*-*	*-*	3.0 ng/mL	0.26 ng/g
	LOQ	1.0 ng/g	1.0 ng/g	0.5 ng/mL	0.01 ng/g
	LOD	0.3 ng/g	0.3 ng/g	0.2 ng/mL	0.003 ng/g
BPE	Range	8.6 (0.5%)	8.1–10.0 (33.3%)	*nd*	*nd*
	Mean	8.6 ng/g	9.0 ng/g	*-*	*-*
	Median	8.6 ng/g	8.6 ng/g	*-*	*-*
	LOQ	2.0 ng/g	2.0 ng/g	0.5 ng/mL	0.10 ng/g
	LOD	0.6 ng/g	0.6 ng/g	0.2 ng/mL	0.03 ng/g

* This value is a CCβ.

**Table 3 animals-12-00410-t003:** The EDCs determined in the 46 samples of raw buffalo milk by HPLC–FLD analysis. The bisphenols and alkylphenols not listed were not detected at all (*nd* = not detected; *>LOD* = detected but not quantified).

Farm	Sample	BPF (ng/g)	BPC (ng/g)	BFDGE (ng/g)	4-t-OP (ng/g)	4-NP (ng/g)
Farm 1	Milk #1	*>LOD*	*nd*	*nd*	*nd*	*nd*
	Milk #4	*nd*	*nd*	*nd*	*>LOD*	*nd*
Farm 2	Milk #2	*nd*	*nd*	*>LOD*	*nd*	*nd*
	Milk #3	*nd*	*nd*	*>LOD*	*nd*	*nd*
	Milk #4	*nd*	*nd*	*>LOD*	*nd*	*nd*
	Milk #5	*nd*	*nd*	*>LOD*	*nd*	*>LOD*
Farm 3	Milk #1	*nd*	*nd*	1.33	*nd*	*nd*
	Milk #2	*nd*	*nd*	*>LOD*	*nd*	*nd*
Farm 4	Milk #1	*nd*	*nd*	*>LOD*	*nd*	*nd*
	Milk #2	*nd*	*nd*	*>LOD*	*nd*	*nd*
	Milk #3	*nd*	*nd*	*>LOD*	*nd*	*nd*
	Milk #4	*nd*	*nd*	*>LOD*	*nd*	*nd*
	Milk #5	*nd*	*nd*	*>LOD*	*nd*	*>LOD*
Farm 5	Milk #1	*nd*	*nd*	*>LOD*	*nd*	*nd*
	Milk #2	*nd*	*nd*	*>LOD*	*nd*	*nd*
	Milk #3	*nd*	*nd*	*>LOD*	*>LOD*	*>LOD*
	Milk #4	*nd*	*nd*	*>LOD*	*nd*	*>LOD*
	Milk #5	*nd*	*nd*	*>LOD*	*nd*	*nd*
Farm 6	Milk #1	*nd*	*nd*	*>LOD*	*nd*	*nd*
	Milk #5	*nd*	*nd*	*>LOD*	*nd*	*nd*
Farm 7	Milk #1	*nd*	*nd*	*>LOD*	*nd*	*nd*
	Milk #2	*nd*	*nd*	*>LOD*	*nd*	*nd*
	Milk #3	*nd*	*nd*	*>LOD*	*nd*	*nd*
	Milk #5	*nd*	*nd*	*>LOD*	*nd*	*>LOD*
Farm 8	Milk #1	*nd*	*>LOD*	*nd*	*nd*	*nd*
	Milk #4	*nd*	*nd*	1.10	*nd*	*>LOD*
	Milk #5	*nd*	*nd*	*>LOD*	*nd*	*nd*
Farm 9	Milk #5	*nd*	*nd*	*>LOD*	*nd*	*nd*
Farm 10	Milk #2	*nd*	*nd*	*>LOD*	*nd*	*nd*
	Milk #3	*nd*	*nd*	*>LOD*	*nd*	*nd*
	Milk #4	*nd*	*nd*	*nd*	1.41	*nd*

**Table 4 animals-12-00410-t004:** The bisphenols determined in 10 samples of retail bovine milk collected from markets in the Campania region and analysed by UHPLC–MS/MS. The other bisphenols were not detected at all (*nd* = not detected; *>LOD* = detected but not quantified).

Sample	Packaging	BPA (ng/mL)	BPF (ng/mL)
Milk #1	Tetra Pak	1.1	10.6
Milk #2	Tetra Pak	0.6	*nd*
Milk #3	Tetra Pak	*>LOD*	*nd*
Milk #4	Tetra Pak	1.3	2.1
Milk #5	Tetra Pak	*nd*	1.5
Milk #6	Tetra Pak	1.3	*nd*
Milk #7	Tetra Pak	*>LOD*	*>LOD*
Milk #8	PET	0.6	0.6
Milk #9	PET	2.8	*>LOD*
Milk #10	PET	0.5	*nd*

**Table 5 animals-12-00410-t005:** The EDCs determined in five samples of retail bovine milk collected from markets in the Lazio region and analysed by HPLC–FLD. All of the bisphenols and alkylphenols not shown in the table were not detected at all (*nd* = not detected; *>LOD* = detected but not quantified).

Sample	Packaging	BPA (ng/g)	BPP + BPM (ng/g)	4-t-OP (ng/g)	4-OP (ng/g)
Milk #1	Tetra Pak	2.19 ± 0.26	*>LOD*	*>LOD*	*nd*
Milk #2	Tetra Pak	3.05 ± 0.37	*>LOD*	*>LOD*	*>LOD*
Milk #3	Tetra Pak	1.11 ± 0.13	*>LOD*	*>LOD*	*>LOD*
Milk #4	Tetra Pak	1.44 ± 0.17	*>LOD*	*nd*	*>LOD*
Milk #5	PET	1.78 ± 0.21	*>LOD*	*>LOD*	*nd*

## Data Availability

Not applicable.

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
