# Peer review of "Determination of 20 Endocrine-Disrupting Compounds in the Buffalo Milk Production Chain and Commercial Bovine Milk by UHPLC–MS/MS and HPLC–FLD"

_animals, 2022, doi:10.3390/ani12040410_

Round 1
Reviewer 1 Report
The authors investigated the occurrence of 20 bisphenols in the buffalo milk production chain and commercial milk collected from Italy by UPLC-MS/MS and HPLC-FLD.This is an innovative work, which contributes to filling the gaps in knowledge regarding the contamination of EDCs in the “feed-cow-milk” production chain. So, this subject can be of interest not only to scientists but also to the general public. However, several critical issues are existing in the manuscript. The manuscript should at least be major revised before publication.
- One of the purposes of this study is to develop the test methods for bisphenols in feed, milk, and serum samples (see lines 68-70). But the authors did not give enough information about the developed methods in the manuscript, especially for sample treatment process (see lines 111-114). I suggest the authors provide more details on test methods development. If authors have published the study on the development of the methods, please give exact literature citations in the text.
- The manuscript is not completed on structure, lacking some analysis of the research background and current situation in bisphenols field (see lines 56-65). It is necessary to review the current progress on bisphenols contamination in the milk production chain and point out the innovation of this study in the part of “introduction”.
- The discussion section is too simple and lacks necessary literature support. I suggest the authors compare the results obtained from this study with some relevant literature data to explain the rationality of the authors` conclusions. Moreover, it is better to provide the reason analysis for the main points of this manuscript.
- Due to the lack of enough discussions, several views are unconvincing. For example, the authors suggested the bisphenols in raw milk was a feed source (see lines 234-239), whereas that in blood was of environmental origin (see lines 240-243). This view seems illogical. It is generally recognized that the raw milk and blood are from cows, and the bisphenols in raw milk and blood are derived from cows' exposure. So, both should have the same bisphenol source. According to my experience, the low detection rates of bisphenols in blood samples might because the test method was not appropriate. Our research has found that most bisphenols were metabolized quickly by animals and existed as II phase metabolites in blood. Thus, for the survey of bisphenols in blood, the samples should be hydrolyzed by Glucuronidase/Arylsulfatase to detect total bisphenols (free- and conjunct-).
- There were manydetail problems, such as but not limited to:
- Lines 58-65, literature cited?
- Line 87, the information of standards and reagents?
- Lines 132-147, the information of RT, MRM and MS/MS parameters?
- Lines 151-187, the matrix effects of bisphenols should be studied, especially for different feed types.
- Lines 151-187, the background levels of bisphenols should be investigated, especially for BPA and BPS, which are ubiquitous in the environment.
- Lines 189-198, the description of the samples needs to be simplified as mentioned above on lines 91-96.
- Lines 214-217, I suggest the authors provide the mean values and median values of the bisphenols in different samples in Table 2.
- Line 230, the authors should provide ajudgment criterion to the conclusion of low contamination levels of BPA in the feed.
- Lines 252-256, the authors should point out the meaning of the numbers on the bar graphs.
- Lines 401-417, the reference literature of No. 18-23 were not marked in the paper.
Author Response
Reviewer #1
Comment #1: One of the purposes of this study is to develop the test methods for bisphenols in feed, milk, and serum samples (see lines 68-70). But the authors did not give enough information about the developed methods in the manuscript, especially for sample treatment process (see lines 111-114). I suggest the authors provide more details on test methods development. If authors have published the study on the development of the methods, please give exact literature citations in the text.
Answer #1: In the revised manuscript we refer to the UHPLC-MS/MS method previously developed and validated in our laboratory. That work is full of data and information about purification and determination of bisphenols, thus was submitted to another journal and is under evaluation; therefore, we cannot provide in the manuscript the additional details requested by the Reviewer, neither we can cite it as a reference now. For these reasons, in the manuscript we report only data of the monitoring, that is the core of the study submitted, but describe the criteria for method validation summarized in subsection 2.4. However, if the Reviewer wants, we could let them view the requested information under non-disclosure condition.
Comment #2: The manuscript is not completed on structure, lacking some analysis of the research background and current situation in bisphenols field (see lines 56-65). It is necessary to review the current progress on bisphenols contamination in the milk production chain and point out the innovation of this study in the part of “introduction”.
Answer #2: A sentence about the state-of-art on the bisphenol contamination in milk production chain and the respective references has been added in the Introduction of the manuscript. The innovation of the study has been pointed out, as suggested by the Reviewer.
Comment #3: The discussion section is too simple and lacks necessary literature support. I suggest the authors compare the results obtained from this study with some relevant literature data to explain the rationality of the authors` conclusions. Moreover, it is better to provide the reason analysis for the main points of this manuscript.
Answer #3: As suggested by the Reviewer, the Discussion section has been improved introducing a comparison between our results and the most relevant literature data on milk and feed contamination from bisphenols.
Comment #4: Due to the lack of enough discussions, several views are unconvincing. For example, the authors suggested the bisphenols in raw milk was a feed source (see lines 234-239), whereas that in blood was of environmental origin (see lines 240-243). This view seems illogical. It is generally recognized that the raw milk and blood are from cows, and the bisphenols in raw milk and blood are derived from cows' exposure. So, both should have the same bisphenol source.
Answer #4: As the Reviewer said, the blood and raw milk contaminations are both consequence of buffaloes’ exposure to bisphenols. However, the buffaloes’ exposure can originate from different sources, not only the dietary intake, for example the soil, when the animals are free and grazing. The presence of the same bisphenols in milk and in feed suggested that the milk contamination had a direct dietary source (from feed to the buffaloes and then to milk). On the other hand, the serum contamination could not originate from dietary source, because the bisphenols detected in blood serum were not present in feed. Therefore, in the case of blood serum, we hypothesized an environmental source of intake, that was not clearly identified (from environment to the buffaloes and then to blood). The sentence has been modified to be clearer.
Comment #5: According to my experience, the low detection rates of bisphenols in blood samples might because the test method was not appropriate. Our research has found that most bisphenols were metabolized quickly by animals and existed as II phase metabolites in blood. Thus, for the survey of bisphenols in blood, the samples should be hydrolyzed by Glucuronidase/Arylsulfatase to detect total bisphenols (free- and conjunct-).
Answer #5: In our laboratory, the serum samples undergo always an enzymatic deconjugation step by β-Glucuronidase/Arylsulfatase. In this case, the sera were previously treated for determination of hematochemical parametrs, then transferred to our laboratory for subsequent analysis of bisphenols. In the revised manuscript we introduced the enzymatic deconjugation step performed by one of the authors.
Comment #6: Lines 58-65, literature cited?
Answer #6: Some relevant references have been introduced, as suggested by the Reviewer
Comment #7: Line 87, the information of standards and reagents?
Answer #7: As suggested by the Reviewer, in the Materials and Method section we introduced information about standards and reagent used during the study. More information are in the manuscript about method development and validation elsewhere submitted. If the Reviewer wants, we could let them view the requested information under non-disclosure condition.
Comment #8: Lines 132-147, the information of RT, MRM and MS/MS parameters?
Answer #8: The detailed information is reported in the work on the method development and validation (as said before). If the Reviewer wants, we could let them view the requested information under non-disclosure condition.
Comment #9: Lines 151-187, the matrix effects of bisphenols should be studied, especially for different feed types.
Answer #9: The matrix effect has been assessed during method development and in subsection 2.4 the use of standard calibration curves both in solvent and matrix-matched is specified. However, the detailed information is reported in the work on the method development and validation (as said before). If the Reviewer wants, we could let them view the requested information under non-disclosure condition.
Comment #10: Lines 151-187, the background levels of bisphenols should be investigated, especially for BPA and BPS, which are ubiquitous in the environment.
Answer #10: The background levels of all bisphenols were assessed during each working session by cleaning up and analysing always process blanks. This way, the possible interference deriving from materials (cartridges, reagents, solvents, glassware, the chromatograph lines) and environment (air, dust) were evaluated. Two sentences have been introduced in the revised manuscript for clarification. Regarding ubiquity of BPA and BPS, it should be always proved. In our previous studies, we observed the contamination is not widespread in all food, when monitored using LC-MS/MS methods. This is a critical point in monitoring; unambiguous identification by mass spectrometry and the elimination of cross contamination by laboratory materials (by process blanks) are mandatory to be avoid false positive contamination.
Comment #11: Lines 189-198, the description of the samples needs to be simplified as mentioned above on lines 91-96.
Answer #11: The manuscript was revised to clarify the statement, as suggested by the Reviewer.
Comment #12: Lines 214-217, I suggest the authors provide the mean values and median values of the bisphenols in different samples in Table 2.
Answer #12: The Table 2 has been revised as suggested by the Reviewer.
Comment #13: Line 230, the authors should provide ajudgment criterion to the conclusion of low contamination levels of BPA in the feed.
Answer #13: In the case of BPA in feed, the 95th percentile of the positive samples was 29.6 ng/g and only in three samples BPA was found at concentration higher than 50.0 ng/g. That is reported in the following part of the subsection 3.1 (“Overall, in about half of the samples […] at levels higher than 100.0 ng/g”). A comparison with literature contamination data has been introduced, to better explain our statement.
Comment #14: Lines 252-256, the authors should point out the meaning of the numbers on the bar graphs.
Answer #14: The meaning of the numbers on the stacked bar graph has been pointed out in the figure caption, as suggested by the Reviewer.
Comment #15: Lines 401-417, the reference literature of No. 18-23 were not marked in the paper.
Answer #15: The references 18-20 have been added in the subsection 2.2, as suggested by the Reviewer. The references 21-23 were already in subsections 2.4 and 3.3.
Reviewer 2 Report
Comments to the Authors
In this work, authors presented the impact of endocrine disrupting compounds (EDC) in the production chain of buffalo milk. Authors looked for 20 EDCs, including bisphenol A, in several matrices throughout the production chain including feed additives, feed, water, milk and serum. They also plan to deepen the survey by analyzing milk-derived products as future perspective.
Two different testing methods were developed and validated to gather quantitative values for investigated EDCs. Results revealed high contamination values in the majority of feed samples and feed additives, while milk and serum samples resulted to be contaminated at low concentration values, and water proved to be EDCs free.
In the end, authors estimated the potential risk to consumers by using contamination values obtained for milk samples and data about consumption of bovine milk reported by WHO.
These results could be interesting to readership, the paper is well written and I think the overall quality is suitable for this journal.
Here are listed some minor issues that should be addressed prior to publication:
- lines 96-97 and 103-104: the sentence is repeated. Please modify.
- lines 100-102: it is difficult to calculate the portion of leachable EDCs without knowing the initial contamination.
- line 98: please change “…cycle step; in this way, …”
- line 113-114: this is just my curiosity: in which material are packed the MIP polymer? And which is the material of the SPE tube? Could it represent a potential source of contamination of blank samples? Same question for QuEChERS (line 134).
- line 115: missing reference, please add the correct referring article.
- line 122: please change “… . For each EDC,…
- line 127: why only two concentration levels instead of three for feed and feed additives? Please support adequately this choice in the manuscript.
- It could be of interest for readers to add a table summarizing details about method validation as supporting table or supplementary material.
- line 136: why two different column chemistry for exactly the same compounds. Hexyl-phenyl for HPLC-MS/MS and C18 for HPLC-FLD?
- line 181: LOD and LOQ for HPLC-FLD method are higher than that of HPLC-MS/MS method. Apart from the possibility to detect 3 additional EDCs using FLD with respect to MS/MS, which is the reason to use FLD as a screening method as proposed in line 74, given that the sample preparation involve an additional step with QuEChERS prior to SPE and the higher LOQ and LOD may increase the risk to have false negative samples?
- lines 214-217: LOD and LOQ values are reported in table 2, but no indications regarding the method (MS/MS or FLD) are indicated in the legend. Please modify.
- lines 240-242: was it possible to look for a correlation among feed samples with serum and/or milk samples coming from the same farm in the same period?
- lines 265-267: this result is a bit unusual especially because MS/MS method possess lower LOQ and LOD with respect to FLD. Could you comment?
- lines 283-285: it is even more difficult without knowing the contamination of milk before the packing.
- lines 338-339: data about the hematochemical parameters appears here, but they were never mentioned in the manuscript.
Author Response
Reviewer #2
Comment #16: lines 96-97 and 103-104: the sentence is repeated. Please modify.
Answer #16: The sentence at lines 103-104 has been deleted, as suggested by the Reviewer.
Comment #17: lines 100-102: it is difficult to calculate the portion of leachable EDCs without knowing the initial contamination.
Answer #17: The retail bovine milk samples were collected and analysed “to evaluate possible consumers’ exposure to EDCs due to environmental pollution or to migration from the packaging”. It was not our aim to pinpoint which one of the two sources is responsible for the EDC contamination in milk. We wanted to quantify the total EDC levels in retail milk samples only to evaluate consumers’ exposure and to compare them with the contamination levels found in raw buffalo milk.
Comment #18: line 98: please change “…cycle step; in this way, …”
Answer #18: the manuscript has been revised as suggested by the Reviewer.
Comment #19: line 113-114: this is just my curiosity: in which material are packed the MIP polymer? And which is the material of the SPE tube? Could it represent a potential source of contamination of blank samples? Same question for QuEChERS (line 134).
Answer #19: The MIP cartridges were in glass or in polypropylene, whereas the SPE tube were in polypropylene; the QuEChERS were purchased as packets of salts and were transferred in glass tubes before sample cleanup. During each working session, a process blank has been cleaned up and analysed to evaluate possible interferences deriving from materials (solvents, reagents, cartridges and glassware) or environment (air, dust). A sentence has been introduced in subsections 2.2 and 2.3 of the manuscript to clarify that.
Comment #20: line 115: missing reference, please add the correct referring article.
Answer #20: The missing references have been added as suggested by the Reviewer.
Comment #21: line 122: please change “… . For each EDC,…
Answer #21: The manuscript has been revised as suggested by the Reviewer.
Comment #22: line 127: why only two concentration levels instead of three for feed and feed additives? Please support adequately this choice in the manuscript.
Answer #22: We validated method for feed at 2 levels according to the procedure of our quality assurance system, because no mandatory requirement is provided by the Regulation (EU) n. 2017/625. From a general point of view, we consider 2 validation levels are sufficient to cover the concentration range of interest
Comment #23: It could be of interest for readers to add a table summarizing details about method validation as supporting table or supplementary material.
Answer #23: We used an UHPLC-MS/MS method previously developed and validated in our laboratory. That work is full of data and information about purification and determination of bisphenols, thus was submitted to another journal and is under evaluation; for this reason, we cannot provide in the manuscript the additional details requested by the Reviewer, neither we can cite it as a reference now. For these reasons, in the manuscript we report only data of the monitoring, that is the core of the study submitted, but describe the criteria for method validation summarized in subsection 2.4. Moreover, the LOQs of the method are provided and discussed. However, if the Reviewer wants, we could let them view the requested information under non-disclosure condition.
Comment #24: line 136: why two different column chemistry for exactly the same compounds. Hexyl-phenyl for HPLC-MS/MS and C18 for HPLC-FLD?
Answer #24: The two test methods were developed and validated independently, by two different partners in a research project funded by the Italian Ministry of Health. Each group worked autonomously, choosing the best chromatographic conditions tested in its own laboratory, using their instrument and chromatographic columns.
Comment #25: line 181: LOD and LOQ for HPLC-FLD method are higher than that of HPLC-MS/MS method. Apart from the possibility to detect 3 additional EDCs using FLD with respect to MS/MS, which is the reason to use FLD as a screening method as proposed in line 74, given that the sample preparation involve an additional step with QuEChERS prior to SPE and the higher LOQ and LOD may increase the risk to have false negative samples?
Answer #25: We agree with the Reviewer observation: the highest LOQ of the HPLC-FLD method would have increased the risk of false negative samples. However, the LOD values of the HPLC-FLD method were at 0.2 ng/g for all the EDCs, except for BPBP that had the LOD at 0.6 ng/g. That allowed us to observe the peak of the EDCs at concentrations comparable to the LOQ of the LC-MS/MS method and, in that case, to confirm their presence. Anyway, all the milk samples were tested with both the analytical methods, regardless of the results of the HPLC-FLD analysis, therefore we are sure there were no false negative samples.
Comment #26: lines 214-217: LOD and LOQ values are reported in table 2, but no indications regarding the method (MS/MS or FLD) are indicated in the legend. Please modify.
Answer #26: The indication about the analytical method has been introduced both in the text of the manuscript and in the table caption, as suggested by the Reviewer.
Comment #27: lines 240-242: was it possible to look for a correlation among feed samples with serum and/or milk samples coming from the same farm in the same period?
Answer #27: There was a time lapse, due to the pandemic onset, between the collection of the feed and milk samples and the collection of blood samples, therefore no correlation was possible among them. Moreover, no correlation was possible even between the contamination levels in feed samples and those in milk samples collected simultaneously, because the sampling was made from the tank gathering milk from all the cows. The data of the monitoring indicate the milk is not heavily contaminated, and this is important for the production of mozzarella cheese. On the other hand, the study of contamination levels in feed gives evidence this is the main source of contamination of milk through feeding.
Comment #28: lines 265-267: this result is a bit unusual especially because MS/MS method possess lower LOQ and LOD with respect to FLD. Could you comment?
Answer #28: BPC has LOD at 0.2 ng/g and LOQ at 1.0 ng/g in the HPLC-FLD method, whereas has LOD at 0.3 ng/ml and LOQ at 1.0 ng/ml in the LC-MS/MS method. As showed in table 3, in the sample “Milk #1” from Farm 8 was found a concentration of BPC, by HPLC-FLD analysis, higher than the LOD but lower than the LOQ. The concentration of BPC was indeed at 0.2 ng/g, that is lower than the LOD value in the LC-MS/MS analysis, therefore the result could not be confirmed.
Comment #29: lines 283-285: it is even more difficult without knowing the contamination of milk before the packing.
Answer #29: See Answer #17. The sentence has been modified to be clearer.
Comment #30: lines 338-339: data about the hematochemical parameters appears here, but they were never mentioned in the manuscript.
Answer #30: A sentence, referring to the hematochemical analysis performed on buffalo blood samples, has been added in the subsection 2.1, as suggested by the Reviewer.